# Comprehensive In Silico Analysis and Transcriptional Profiles Highlight the Importance of Mitochondrial Dicarboxylate Carriers (DICs) on Hypoxia Response in Both *Arabidopsis thaliana* and *Eucalyptus grandis*

**DOI:** 10.3390/plants11020181

**Published:** 2022-01-11

**Authors:** Pedro Barreto, Mariana L. C. Arcuri, Rômulo Pedro Macêdo Lima, Celso Luis Marino, Ivan G. Maia

**Affiliations:** Department of Chemical and Biological Sciences, Institute of Biosciences of Botucatu, UNESP, Botucatu 18618-689, Brazil; pedropbarretto@gmail.com (P.B.); mariana.arcuri@unesp.br (M.L.C.A.); romulo.lima@unesp.br (R.P.M.L.); cl.marino@unesp.br (C.L.M.)

**Keywords:** dicarboxylates, DIC, mitochondria, abiotic stress, hypoxia, submergence

## Abstract

Plant dicarboxylate carriers (DICs) transport a wide range of dicarboxylates across the mitochondrial inner membrane. The *Arabidopsis thaliana*
*DIC* family is composed of three genes (*AtDIC1*, *2* and *3*), whereas two genes (*EgDIC1* and *EgDIC2*) have been retrieved in *Eucalyptus grandis*. Here, by combining in silico and *in planta* analyses, we provide evidence that DICs are partially redundant, important in plant adaptation to environmental stresses and part of a low-oxygen response in both species. *AtDIC1* and *AtDIC2* are present in most plant species and have very similar gene structure, developmental expression patterns and absolute expression across natural Arabidopsis accessions. In contrast, *AtDIC3* seems to be an early genome acquisition found in Brassicaceae and shows relatively low (or no) expression across these accessions. In silico analysis revealed that both *AtDICs* and *EgDICs* are highly responsive to stresses, especially to cold and submergence, while their promoters are enriched for stress-responsive transcription factors binding sites. The expression of *AtDIC1* and *AtDIC2* is highly correlated across natural accessions and in response to stresses, while no correlation was found for *AtDIC3*. Gene ontology enrichment analysis suggests a role for *AtDIC1* and *AtDIC2* in response to hypoxia, and for *AtDIC3* in phosphate starvation. Accordingly, the investigated genes are induced by submergence stress in *A. thaliana* and *E. grandis* while *AtDIC2* overexpression improved seedling survival to submergence. Interestingly, the induction of *AtDIC1* and *AtDIC2* is abrogated in the *erfVII* mutant that is devoid of plant oxygen sensing, suggesting that these genes are part of a conserved hypoxia response in Arabidopsis.

## 1. Introduction

Dicarboxylates are required in essential plant metabolic pathways including the metabolism of fatty acids, the synthesis of amino acids and gluconeogenesis [1]. These compounds have also been implicated in the transfer of reducing equivalents for photorespiration [2,3]. The transport of dicarboxylic acids across the inner mitochondrial membrane is commonly mediated by mitochondrial dicarboxylate carriers (DIC) that are members of the mitochondrial carrier family (SLC25) [4,5]. Three plant DIC homologues (namely *AtDIC1*, *AtDIC2* and *AtDIC3*) showing distinct substrate specificities with regard to their yeast and animal counterparts were identified in the model plant *Arabidopsis thaliana* [3]. AtDIC1 and AtDIC2 are closely related (70% of identical amino acid residues) and more abundantly expressed than AtDIC3, which shares 55–60% amino acid identity with the other isoforms. *AtDIC3* also shows a restricted pattern of expression (flowers and siliques) compared to the almost ubiquitous expression of *AtDIC1* and *AtDIC2* [3]. Recombinant AtDIC proteins reconstituted into liposomes have been shown to exchange sulfate with malate, oxaloacetate, phosphate and succinate, and to a lesser extent 2-oxoglutarate [3].

Data from publicly available expression databases reveals that the transcripts of *AtDIC1* and *AtDIC2* accumulate in response to different environmental cues including exposure to cold, drought, phosphate limitation and UV radiation [6]. In line, *AtDIC1* figured among the most stress-responsive genes encoding mitochondrial proteins identified [7], while a rapid and potent induction of the *AtDIC* genes in response to touch-stimulated mechanical stress has been reported [8,9]. The transcriptional up-regulation elicited by touch was proposed to address a physiological change since a significant reduction in DIC substrates such as succinate and citrate was observed in treated plants [9]. In this context, AtDIC2 was recently shown to promote the export of cytosolic malate in exchange for mitochondrial citrate [10], an activity that contributes for the maintenance of metabolic homeostasis, especially under stressful conditions. Interestingly, mutant plants lacking *AtDIC2* were impaired in growth [10], a phenotype that further supports the importance of this carrier to normal metabolic function. Collectively, these findings suggest that modulation of DIC activity and expression might be part of the strategy of plant metabolic reprogramming required to overcome adverse growth conditions.

Among different environmental clues, hypoxic stress is known to elicit important changes in cellular metabolism by promoting glycolysis and fermentation pathways [11,12]. Although influenced by different factors, the main metabolic responses to oxygen deficiency involve alterations in the levels of tricarboxylic acid cycle (TCA) intermediates, including the accumulation of succinate and malate. In this regard, given their transport characteristics, DICs have been suggested to contribute to the metabolic adjustments required for plant adaptation to low-oxygen conditions, a feature that is supported by the reported induction of *AtDIC2* under hypoxia and anoxia [13].

Despite the reported information, knowledge regarding the function of DICs in plants is still limited. Here, by combining in silico and in vivo analysis, we provide evidence that DICs are partially redundant, important in plant adaptation to environmental stresses and part of a low-oxygen response in both *A. thaliana* and *Eucalyptus grandis*, an economically important forestry species. It should be emphasized that eucalyptus plantations in South America have been facing problems associated with short-term hypoxia mainly due to soil compaction [14] and a better understanding of the underlying molecular mechanisms implicated in adaptation to hypoxia in woody plants is of special interest.

## 2. Results and Discussion

### 2.1. Dicarboxylate Carriers Are Widely Distributed in the Plant Kingdom

The Arabidopsis DIC sequences (*AtDIC1*-*3*) were used as queries to search for the presence of DICs in different species across the plant kingdom. We could find DICs orthologues, inferred by a bi-directional best hit approach, in Chlorophyta, Embryopyite and Tracheophyte phyla in addition to early angiosperms such as *Amborella trichopoda* (Appendix A). We also searched for DICs in several higher plant species including *E. grandis* (Appendix A). Two genes encoding DICs (hereafter named *EgDIC1* and *EgDIC2*) were identified in the *E. grandis* genome (Figure 1a). While *EgDIC1* is located on chromosome 9 and has a single transcript variant, *EgDIC2* is located on chromosome 6 and encodes two distinct transcript variants (*EgDIC2 X1* and *X2*) (Figure 1b). The main difference between these two variants is the presence of a 5′-untranslated region (UTR) in *EgDIC2 X1*. The presence of introns was exclusively found in *AtDIC3* and *EgDIC2* (Figure 1b). Intriguingly, eight homologues belonging to the so called *AtDIC2* subgroup have been annotated in *Populus trichocarpa* [15].

Amino acid sequence alignments revealed that the predicted *E. grandis* proteins are apparently more closely related to AtDIC1 (Appendix A). Interestingly, the majority of the DIC orthologues that we found across plant species, including EgDIC1 and EgDIC2, are annotated as Mitochondrial Uncoupling Protein-like or Mitochondrial Uncoupling Protein (UCP). This is mainly because DICs were previously thought to be part of the UCP family due to the high sequence homology among them [16]. It has been demonstrated, however, that recombinant and purified AtDICs transport the characteristic substrates of dicarboxylate carriers by an exchange mechanism [3], whereas UCPs may transport protons and/or amino acids [17] (Figure 1c). In this context, our phylogenetic analysis clearly shows that plant UCP1 orthologues and DICs form two distinct groups (Figure 1d). Interestingly, EgDIC1 is more closely related to AtDIC1, while both variants of EgDIC2 cluster together with AtDIC2. On the other hand, although being clearly distinct from UCPs, AtDIC3 is phylogenetically distant from other DICs. In this regard, AtDIC3 forms a cluster with *Oryza sativa* UCP5 (OsUCP5) and seems to be a close orthologue of *Brassica napus* UCP6-like (BnUCP6-like). By using a bi-directional best hit analysis, we confirmed that AtDIC3 is a true orthologue of BnUC6-like, but this is not the case for OsUCP5 (Appendix A). True orthologues of AtDIC3 were also found in several species of Brassicaceae such as *Arabidopsis lyrata*, *Brassica oleracea*, *Capsella rubela* and *Camelina sativa* (Appendix A). In addition, we performed an extensive search in public genomic sequences available at Phytozome [18] and could not find AtDIC3 orthologues in any species outside the Brassicaceae, thus suggesting that this gene is an early genomic acquisition of this family during the course of evolution.

### 2.2. Analysis of Cis-Regulatory Elements in DIC Promoters

The promoter sequences (1.5 kb upstream from the AUG start codon) of each *A. thaliana* and *E. grandis DIC* genes were used as queries in PlantCare software [19] for the identification of *cis*-acting regulatory elements. This allowed us to depict the frequency of occurrence of different *cis*-elements accordingly to their position in the corresponding promoter regions using both reverse and forward strands (Figure 2). The frequency distribution varied greatly among *A. thaliana* and *E. grandis*. The higher frequency of binding sites was found between −800 to −1000 base pairs (bp) and −1000 to −1400 bp from the translation initiation site in the promoter regions of *AtDIC1* (Figure 2a) and *AtDIC2* (Figure 2b), respectively. On the other hand, the high frequency of regulatory elements found in the promoter of *AtDIC3* lies between −200 to −400 bp upstream of the translation initiation site (Figure 2c). Interestingly, the frequency of binding sites at the first 200 bp upstream from the AUG is very low in *AtDIC3* when compared to the other *AtDICs*. Predicted binding sites for different transcription factors (TF) were evenly distributed along the upstream regions of *EgDICs* (Figure 2d,e). In this context, *AtDIC2* (48) and *EgDIC2* (47) have the higher number of distinct *cis*-regulatory elements in their promoters. The functional categorization of these promoter elements revealed that responsiveness to light, abscisic acid (ABA) and hypoxia are common features of *A. thaliana* and *E. grandis DIC* genes (Appendix A).

The identified *cis*-regulatory elements were subsequently grouped into their corresponding TF families using AGRIS [20] and PlantPan [21] databases for *A. thaliana* and *E. grandis*, respectively. Remarkably, we found an over-representation of binding sites for AT-HOOK containing TFs in the promoter regions of *AtDICs* (Figure 2a–c). Almost 40% of the *cis*-regulatory elements found in the *AtDIC1* promoter, for example, contained AT-HOOK binding motifs, which are recognized by AT-hook motif nuclear localized (AHL) TFs. AT-hook motif proteins are chromatin modification proteins that participate in a wide array of cellular processes in mammals, including DNA replication, repair and transcription [22]. The Arabidopsis genome encodes 29 AHL proteins that also contain a conserved domain (namely PPC) that directs nuclear localization and contributes to the interaction of AHL with other nuclear proteins [23]. Over the past few years, members of the AHL family have been implicated in plant stress responses and immunity [24,25] and in the regulation of developmental processes such as hypocotyl growth [26], floral transition and development [27,28,29], senescence [30] and vascular tissue patterning [23].

On the other hand, the promoter regions of *EgDICs* harbored *cis*-regulatory elements recognized by different TF families, the most representative ones being AP2 (18.13%) and Myb/SANT (17.58%) in *EgDIC1* and bHLH (19.05%) and bZip (17.05%) in *EgDIC2* (Figure 2d,e). These observations suggest that these isoforms might play distinct roles in *E. grandis*. There are few studies on the function of TFs in eucalyptus. It has been shown, for example, that *EgMyb1* overexpression alters vascular development resulting in fewer lignified fibers and reduced secondary wall thickening in both *A. thaliana* and *P. trichocarpa* [31]. Moreover, an extensive characterization of the *E. grandis AP2/ERF* (APETALA2/Ethylene-Responsive element binding Factor) family of TFs revealed an intriguing over-representation of members of the stress-responsive *DREB* (Dehydration Responsive Element Binding) subfamily, more specifically of *DREB1/CBF* and *DREB2* genes [32]. There are a large number of studies that report the responsiveness of these genes to various abiotic stresses. In this context, while *CBF* is reported to participate mainly in Arabidopsis response to low temperature [33], *DREB* is implicated in drought stress responses in several plant species [34]. Surprisingly, the expression profiles of *CBF* and *DREBs* in *E. grandis*, with regard to other species, revealed a strong responsiveness to heat stress [32]. On the other hand, members of the bHLH (basic helix-loop-helix) and bZIP (basic leucine zipper) TF families have been studied in Poplar revealing a role for bHLH members in drought stress tolerance [35], whereas 45 out of the 96 bZIP coding genes present in the poplar genome are responsive to salt stress [36].

In order to gain further insights into possible regulators of *AtDICs* expression, we first used the Genevestigator database to search for genes that are co-expressed with *AtDICs* in a wide range of perturbations. In parallel, a similar analysis was conducted using the data from the Arabidopsis 1001 Genomes Project [37], which represents a valuable tool to investigate natural variation among genes. The co-expressed transcripts were used as queries against the Plant Transcription Factor Database [38] to obtain a list of co-expressed TFs. By combining both datasets, we discovered several potential regulators of *AtDICs* expression (Figure 2f–i). According to the Genevestigator data, we found 14 TFs that are co-expressed with *AtDIC1*, while 29 TFs were found as co-expressed using the Arabidopsis 1001 Genomes Project. Interestingly, 11 TFs were in common between the two conducted analyses (Figure 2f). The same procedure was employed for the analysis of *AtDIC2* and *AtDIC3*. By associating the data obtained from Genevestigator and the 1001 Arabidopsis Genomes, we identified 21 TFs that are co-expressed with *AtDIC2* (Figure 2g). In contrast, no TFs were found co-expressed with *AtDIC3* in the Arabidopsis 1001 Genomes dataset, whereas 10 were found using the Genevestigator dataset (Figure 2h). Interestingly, four TFs were found to be co-expressed with all *AtDIC* genes, while 6 were in common between *AtDIC1* and *AtDIC2*, two between *AtDIC2* and *AtDIC3* and none between *AtDIC1* and *AtDIC3* (Figure 2i; Appendix A). These four TFs that are co-expressed among all *AtDICs* include AT1G80840 (WRKY), AT3G23250 (MYB), AT1G28370 (ERF) and AT1G27730 (C2H2) that are all associated with abiotic and biotic stress responses. Interestingly, the *Hypoxia Response Factor 2* (HRE2, AT2G47520), a member of the ERF family, was co-expressed with both *AtDIC1* and *AtDIC2* (Appendix A). This TF is both part of the plant low oxygen response [39] and a member of the mitochondrial dysfunction stimulon [40]. Overall, these results confirm the presence of several stress-responsive TF-binding sites in the investigated gene promoters. Moreover, the co-expression analysis suggests that *AtDIC* expression might be co-regulated by several transcription factors during exposure to biotic and abiotic stresses.

### 2.3. Functional Insights from In Silico Gene Expression Analysis

We searched public transcriptomic datasets using Genevestigator software to investigate the expression patterns of *A. thaliana DICs* along plant development. In contrast to *AtDIC2* and *AtDIC3*, transcripts of *AtDIC1* are already present in seeds (Figure 3a). Subsequently, as plants go through the seedling stage, *AtDIC2* is induced by at least 4-fold when compared to seeds. During the following developmental stages, the expression patterns of *AtDIC1* and *AtDIC2* are very similar, remaining constant until the developed flower stage, in which an expression peak is observed for both genes. In contrast, when compared to the other *AtDICs*, *AtDIC3* expression remains very low in all developmental stages. In this regard, although showing in vitro the transport characteristics of other DICs [3], the relatively low expression levels of *AtDIC3* in plant tissues argue against a pivotal role for this isoform in *A. thaliana* development. To further investigate this possibility, we used the transcriptome datasets from 727 distinct *A. thaliana* ecotypes provided by the Arabidopsis 1001 Genomes Project. The average expression of *AtDIC1* (TPM = 2076) and *AtDIC2* (TPM = 2163) was found to be very similar among these accessions, while *AtDIC3* presented very low average expression values (TPM = 39) (Figure 3b). Interestingly, *AtDIC3* expression was absent in at least 25 of the investigated accessions and its highest expression value was 1003, less than half of the detected average expression of *AtDIC1* and *AtDIC2*. In line with these data, a single organelle proteome study retrieved 51 and 29 copies of AtDIC1 and AtDIC2, respectively, per mitochondria, while no copy of AtDIC3 was found [41]. It seems therefore that AtDIC3 activity is required under specific circumstances and at low levels.

In contrast to *A. thaliana*, publicly available transcriptomic data for *E. grandis* is sparse. By searching in the EUCANEXT database [42], we found that *EgDIC1* has similar expression in roots, stem and leaves, whereas *EgDIC2* expression is around 2.5-fold higher in roots and stem when compared to leaves (Figure 3c). We could also observe that *EgDIC1* expression was almost 2 times higher in the xylem of *E. globulus* when compared to *E. urophyla* and *E. grandis*, whereas *EgDIC2* transcript abundance was higher in *E. urophyla* (Figure 3d).

In order to gain further insights into the functionality of the investigated *AtDICs*, we evaluated their expression patterns across a wide range of *A. thaliana* perturbations and in response to phytohormones (Figure 4a–c). *AtDIC1* was found to be strongly upregulated in plants subjected to ozone (32-fold), *Pseudomonas syringae* (22-fold), high light (18-fold), cold (15-fold), drought (14-fold) and in whole seedlings subjected to hypoxia (4-fold) (Figure 4a). With the exception of *P. syringae, AtDIC2* showed similar responsiveness to the aforementioned treatments, but with detected differences in the magnitude of induction (Figure 4b). Moreover, this gene was also highly induced by treatment with the protein synthesis inhibitor cycloheximide. Interestingly, *AtDIC1* and *AtDIC2* were downregulated in the roots of seedlings subjected to anoxia (4-fold) and hypoxia (4-fold), respectively (Figure 4a,b). On the other hand, apart from being induced by cycloheximide and cold, *AtDIC3* expression was also upregulated by phosphate deficiency (13-fold), antimycin (13-fold) and ABA (5-fold) (Figure 4c). A similar analysis conducted using the BAR database [43] confirmed the responsiveness of *AtDIC1* and *AtDIC2* to cold, osmotic and drought stresses (Figure 4d). *AtDIC3* expression did not differ by more than 2-fold in magnitude among most of the above-mentioned treatments, although it was significantly repressed after 3 h of drought stress by more than 3-fold. Absolute data and statistics for Figure 4d are available at Appendix A. Likewise, all three *AtDICs* were induced after exposure to biotic stresses (Figure 4e). In this regard, infection with *Botrytis cinerea* provoked a more than 4-fold increase in *AtDIC1* and *AtDIC3* expression, while all three *AtDICs* were significantly upregulated after treatment with the bacterial elicitors HrpZ or Flg22 (Figure 4e).

When looking at the EUCANEXT database, we found that *EgDIC1* expression in roots was reduced by cold (65%) and frost (35%) treatments (Figure 4f). A similar pattern was found in leaves, where *EgDIC1* expression was repressed by 80% and 60% after cold and frost treatment, respectively. A negative impact in *EgDIC2* expression in roots in response to cold (77%) and frost (36%) was also detected (Figure 4f). Intriguingly, *EgDIC2* expression was not altered by these stresses in leaves. The retrieved data also shows that plants grown under different environmental conditions (humid, semi-arid and dry) presented about the same amount of *EgDICs* transcripts (Figure 4g). An exception was found in leaves from *E. grandis* grown in dry environments, in which a 2-fold increase in *EgDIC2* transcript abundance was detected.

### 2.4. Natural Variation of DICs Expression Reveals a High Correlation between DIC1 and DIC2

We sought to predict whether the similar expression behavior of *AtDIC1* and *AtDIC2* during plant development, in response to stresses and in distinct accessions, could be due to co-expression. For that, we calculated the Pearson correlation coefficient (r) using the data from the Arabidopsis 1001 Genomes Project. Notably, we found that *AtDIC1* and *AtDIC2* are highly co-expressed in the investigated accessions (Figure 5a). The dispersion of *AtDIC2* expression strongly resembles the one observed for *AtDIC1* (Figure 5a), resulting in a high positive correlation between them (r = 0.63) (Figure 5b). In contrast, *AtDIC3* behaved very differently and no correlation with *AtDIC1* (r = 0.09) nor *AtDIC2* (r = 0.03) could be found (Figure 5b).

A GO enrichment analysis was subsequently performed to determine the biological processes terms associated with the genes presenting elevated correlation coefficients with *AtDICs* (Figure 5c). It should be emphasized that *AtDIC1* and *AtDIC2* were highly correlated with 113 and 91 transcripts, respectively (r ≥ 0.75) (Appendix A). These transcripts were significantly enriched in several GO terms, being the top 2, “response to chitin” and “response to hypoxia” (Figure 5c). In contrast, very few transcripts were correlated with *AtDIC3* (Appendix A), a feature that resulted in a small number of GO categories found enriched, i.e., galactolipid biosynthetic processes, response to phosphate starvation and transmembrane transport. It is known that genes that are functionally related are often tightly connected and the co-expression links are frequently conserved across plant networks [44]. These results suggest that AtDIC1 and AtDIC2 may be partially redundant, complementary or act synergistically during *A. thaliana* development and, specially, in response to stresses.

### 2.5. DICs Are Responsive to Submergence in A. thaliana and E. grandis

According to our in silico analysis, submergence and hypoxia are among the major perturbations that can alter *AtDICs* expression (Figure 4a,b). In agreement, *AtDIC2* was previously reported to be induced by hypoxia and anoxia [13]. Moreover, several transcripts associated with the term “response to hypoxia” were found co-expressed with *AtDIC1* and *AtDIC2* (Figure 5c). In view of such evidence, we decided to investigate whether DICs could be also part of the submergence response in *E. grandis* by waterlogging 2-month-old seedlings (Figure 6a). Seedling biomass, both in terms of Fresh Weight (FW) and Dry Weight (DW), together with Relative Water Content (RWC), were strongly affected by 30 days of waterlogging stress (Figure 6b). Previous reports in waterlogged seedlings of gray poplar (*Populus × canescens*) revealed that most of the changes in hypoxia-related transcripts occur in the roots rather than leaves [45]. In fact, we found that both *EgDIC1* and *EgDIC2* are strongly responsive to short term submergence stress in *E. grandis* roots (Figure 6c). *EgDIC1* transcript levels in roots increased 28-fold and 31-fold after 24 h and 48 h of waterlogging. Similarly, *EgDIC2* transcript abundance increased approximately 10-fold and 20-fold after 24 h and 48 h under waterlogging conditions. On the other hand, the expression of both genes returned to basal levels after 72 h of waterlogging. Interestingly, hypoxia has been shown to negatively affect the levels of 2-oxoglutarate, a pivotal metabolite of the TCA cycle, in the roots of a *Eucalyptus urograndis* clone tolerant to oxygen depletion, whereas a significant decrease in citrate levels was observed in the roots of a sensitive clone [14]. In addition, an increase in gamma-aminobutyric acid (GABA) and succinate (2.5-fold) levels were observed in the roots of both clones under hypoxic conditions [14]. These data suggest that modulation of the TCA cycle intermediates is part of the *Eucalyptus* metabolic adjustments during exposure to short-term hypoxia, a feature that probably requires DIC activity.

Hypoxia response in *A. thaliana* is regulated by the Group VII Ethylene Transcription Factors (ERFVIIs) [39,46,47,48]. In order to investigate whether ERFVIIs might control *AtDICs* expression under submergence, we employed the quintuple Arabidopsis mutant *rap2.12 rap2.2 rap2.3 hre1 hre2* (*erfVII*) that lacks hypoxia response. In seedlings of the Col-0 background, both *AtDIC1* and *AtDIC2* are induced shortly after 10 min of submergence stress, reaching their peak of expression after 30 min (Figure 6d). In contrast, the observed induction is abrogated in the *erfVII* mutant, suggesting that both *AtDIC1* and *AtDIC2* could be under the control of ERFVIIs, at least during submergence. On the other hand, *AtDIC3* expression is not induced in submerged seedlings of both genotypes.

To provide further support for the involvement of DICs in the hypoxia response, we employed independent Arabidopsis transgenic lines overexpressing *AtDIC2* (Appendix A; E#1, E#8 and E#9). It should be emphasized that, according to our results, *AtDIC2* and *AtDIC1* are responsive to submergence in an ERFVII-dependent manner (Figure 6d). Therefore, a survival assay by submerging seedlings in H_2_O for 60 min in the dark was performed. The results show that *AtDIC2* overexpression increased seedling survival to submergence compared with the wild-type Col-0 (Figure 6e). As expected, the *prt6* line, which has a constitutive hypoxia response due to the lack of PRT6 E3 ligase activity that targets N-end rule pathway substrates for proteasomal degradation [39], also displayed improved seedling survival. In contrast, the *erfVII* mutant was clearly more susceptible to this stress (Figure 6e). The fact that *AtDIC2* overexpression is able to confer tolerance to submergence is indicative of a critical role for DICs during low-oxygen response.

It is known that transcripts encoding TCA cycle enzymes, such as Malate Dehydrogenase, are down-regulated in response to low-oxygen in poplar and Arabidopsis [49]. In agreement with that, several of their corresponding metabolites decrease in abundance under anoxia in rice [12]. A postulated role played by DICs is linked to the metabolic flux of organic acids to or from the mitochondria [3]. In this scenario, *AtDICs* might be important to export mitochondrial dicarboxylates, especially malate, to the cytosol where it could be oxidized to oxaloacetate and NADH, providing reducing equivalents for ethanolic fermentation [16]. Taken together, these results suggest that *AtDIC1* and *AtDIC2* in *A. thaliana* and *EgDIC1* and *EgDIC2* in *E. grandis* are part of the low-oxygen response. As *EgDICs* are responsive to waterlogging, we hypothesize that these genes might be also under control of EgERFVIIs and wonder whether this mechanism might be extrapolated to angiosperms.

## 3. Material and Methods

### 3.1. Experimental Model and Subject Details

The *prt6* and *erfvii* mutant line of *Arabidopsis thaliana* was previously described [39,46,47,48] and kindly provided by Prof. Michael Holdsworth from University of Nottingham. The Col-0 accession was used as wild-type (WT) control in the assays. *Arabidopsis thaliana* (Col-0) lines overexpressing *AtDIC2* under the control of the *CaMV* 35S promoter were generated using the floral dip method essentially as described [40]. For that, the full-length cDNA of *AtDIC2* was PCR-amplified using gene-specific primers (AtDIC2c; Appendix A) and cloned into the pBI121 binary vector digested by *BamHI* and *SacI*. Transformants were selected on half strength MS media containing 100 μg/mL kanamycin. Three independent transgenic lines (T2) showing distinct levels of *AtDIC2* expression (namely E#1, E#8 and E#9; Appendix A) were employed in the submergence assays. A publicly available clone of *Eucalyptus grandis* (G00510) was employed in all assays.

### 3.2. Phylogenetic Analysis

The amino acid sequences of *A. thaliana* (At) DIC1 (NP_179836.1), DIC2 (NP_194188.1) and DIC3 (NP_196509.1) were used as queries for searching DICs in a number of plant species employing BLASTP [50]. Multiple amino acid sequence alignments of DICs and other mitochondrial proteins were performed using MUSCLE [51]. The phylogenetic analyses were conducted using the maximum likelihood algorithm corrected by Poisson distribution on MEGA 7 [52] and tested by the bootstrap method with 1000 replications. Accession numbers for proteins used in the phylogenetic analysis are available at Appendix A.

### 3.3. In Silico Gene Expression and Promoter Analysis

*Arabidopsis thaliana* gene expression analysis was performed using both Affymetrix and mRNA-seq libraries from Genevestigator (https://genevestigator.com/gv/ (accessed on 1 May 2021) and the Bio-Array Resource [43]. The Arabidopsis 1001 Genomes [37] population data were used for analyzing the expression and correlation among *AtDIC* family members in distinct Arabidopsis ecotypes. Genevestigator and the Arabidopsis 1001 Genomes databases were also used for the co-expression analyses. A list of co-expressed TFs was obtained by submitting co-expressed transcripts as queries against the Plant Transcription Factor Database [38]. The EUCANEXT [42] database was used to investigate *EgDICs* expression patterns in *E. grandis*.

Promoter sequences (1.5 kb upstream of translation start site) were retrieved for each *A. thaliana* and *E. grandis* DIC from NCBI. The tools PlantCare (http://bioinformatics.psb.ugent.be/webtools/plantcare/html/ (accessed on 15 May 2021) [19], AGRIS (http://arabidopsis.med.ohio-state.edu/AtcisDB/ (accessed on 15 May 2021) [20] and PlantPan (http://plantpan.itps.ncku.edu.tw/ (accessed on 15 May 2021) [21] were employed for scanning for *cis*-elements present in the promoter regions of the investigated genes. The identified *cis*-elements were compared with each other and discussed in light of literature available.

### 3.4. Submergence Assays

Seeds from the *erfvii* mutant line and WT Col-0 were sowed on half-strength MS medium pH 5.7 containing 1% (*w/v*) agar, stratified for 96 h at 4 °C and grown in vertical plates at 16 h light conditions for 7 days. Approximately 30 seedlings were submerged in H_2_O in a 1.5 mL tube for 0, 10, 30 and 60 min in the dark. Seedlings were removed from the tube and snap-frozen in liquid nitrogen. The same number of seedlings was maintained in a humidity chamber for 0, 10, 30 and 60 min in the dark as controls. Two independent experiments were carried out.

The survival of Arabidopsis seedlings to submergence was determined as previously described [53] with modifications. Seeds were sown on quarter-strength MS medium pH 5.7 containing 1% (*w/v*) agar, stratified for 96 h at 4 °C and grown in vertical plates at neutral day conditions for 4 days. Seedlings were then submerged in sterile H_2_O in a 1.5 mL tube for 60 min in the dark and returned to quarter-strength MS in the growth chamber. After 4 days, the root tips were scored based on survival (continued growth of the primary root). Control plants were submerged in the light for the same amount of time as treated plants and returned to quarter-strength MS in the growth chamber. Three independent experiments were carried out. Each experiment was conducted using 30 seedlings for each tested genotype.

For the submergence assays in *E. grandis*, two-month-old seedlings (clone G00510) were transferred to 0.5 L pots filled with a mixture of soil-vermiculite (2:1), maintained for 2 weeks in a 16 h light regime at 27 °C for acclimatization, and then placed inside large plastic containers. Submergence was imposed by filling the containers with water until 2 cm above soil. Submerged samples were collected after 0, 24, 48, 72 and 240 h, washed in H_2_O and dried with a paper towel. Samples were then snap-frozen in liquid nitrogen and stored at −80 °C. Fresh weight (FW) and dry weight (DW) of aboveground biomass were measured using the seedlings subjected to 30 days of waterlogging and used to calculate the relative water content (RWC) as described [54]. A total of 12 biological replicates were used to evaluate FW, DW and RWC.

### 3.5. Gene Expression Analysis

Total RNA was isolated from Arabidopsis seedlings using a RNAeasy Plant Mini Kit (Qiagen, Hilden, Germany), whereas a CTAB derived extraction protocol followed by LiCl precipitation with minor modifications was used for isolation of total RNA from Eucalyptus roots [55,56]. One microgram of the extracted RNA samples was treated with RNase-free DNase I (Promega, Madison, WI, USA) and then reverse-transcribed using a High-Capacity cDNA Synthesis kit (Thermo Fisher, Waltham, MA, USA) and random primers according to the manufacturer’s protocol.

Real-time PCR was performed using the Applied StepOne system (Applied Biosystems, Foster City, CA, USA) and carried out in a final volume of 10 μL using the following cycling conditions: initial denaturation at 95 °C for 5 min, followed by 45 cycles of 95 °C for 15 s and 60 °C for 1 min. Each qPCR reaction consisted of 6 µL of GoTaq Colorless qPCR Master Mix (Promega, Madison, WI, USA), 1 µL of cDNA (1/10 dilution) and 0.3 µM of forward and reverse primers, respectively. The reactions were performed in technical triplicates with at least two biological replicates. The results were expressed relative to the expression levels of 40S and TUBULIN genes (for Arabidopsis and *Eucalyptus*, respectively) using the 2–ΔΔCt method. All values were expressed as fold changes of treated plants relative to the control. A list of the primers used in this study is available at Appendix A.

### 3.6. Statistical Analysis

Results from Real-time PCR were analyzed using Student’s *t*-test to determine statistical significance (* *p* < 0.05, ** *p* < 0.01). Data obtained from Genevestigator were previously filtered using a 2-fold change cutoff and *p* < 0.01. The statistical significance of data obtained from BAR was determined using a one-way ANOVA and mean values were compared using Tukey’s test.

## Figures and Tables

**Figure 1 plants-11-00181-f001:**
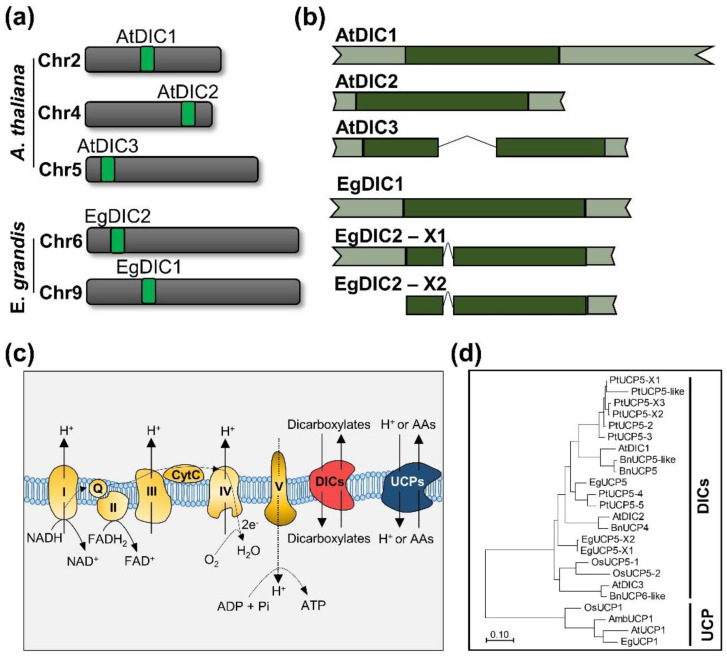
DICs are a multigenic family of dycarboxylate carriers widely distributed in plants. Chromosome location (**a**) and gene structure (**b**) of *Arabidopsis thaliana* and *Eucalyptus grandis* DICs. (**c**) DICs transport dicarboxylates across the mitochondrial inner membrane [3] while UCP1 transport protons or amino acids [17]. (**d**) Molecular phylogenetic analysis by maximum likelihood method (consensus of 1000 bootstrap replicates) of DICs and UCP1 from different plant species. Pt: *Populus trichocarpa,* At: *A. thaliana*, Bn: *Brassica napus*, Eg: *E. grandis*, Os: *Oryza sativa*, Amb: *Amborella trichopoda*. The scale bar represents 0.1 amino acid substitution per site.

**Figure 2 plants-11-00181-f002:**
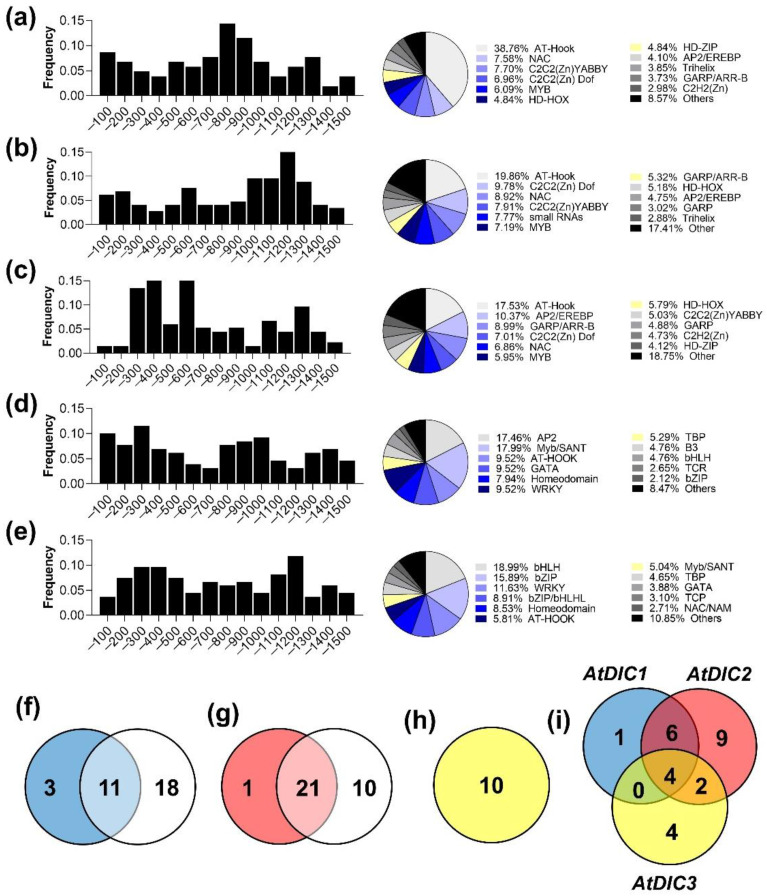
Promoter and co-expression analysis reveal transcription factors that are potential regulators of *AtDICs* expression. Frequency distribution of transcription factors (TF) binding sites and categorization of the corresponding TF families observed in the promoter regions of (**a**) *AtDIC1*, (**b**) *AtDIC2*, (**c**) *AtDIC3*, (**d**) *EgDIC1* and (**e**) *EgDIC2.* Venn diagrams showing the number of TFs co-expressed with (**f**) *AtDIC1*, (**g**) *AtDIC2* and (**h**) *AtDIC3*. (**i**) Venn diagrams showing the number of TFs that are co-expressed among *AtDICs*.

**Figure 3 plants-11-00181-f003:**
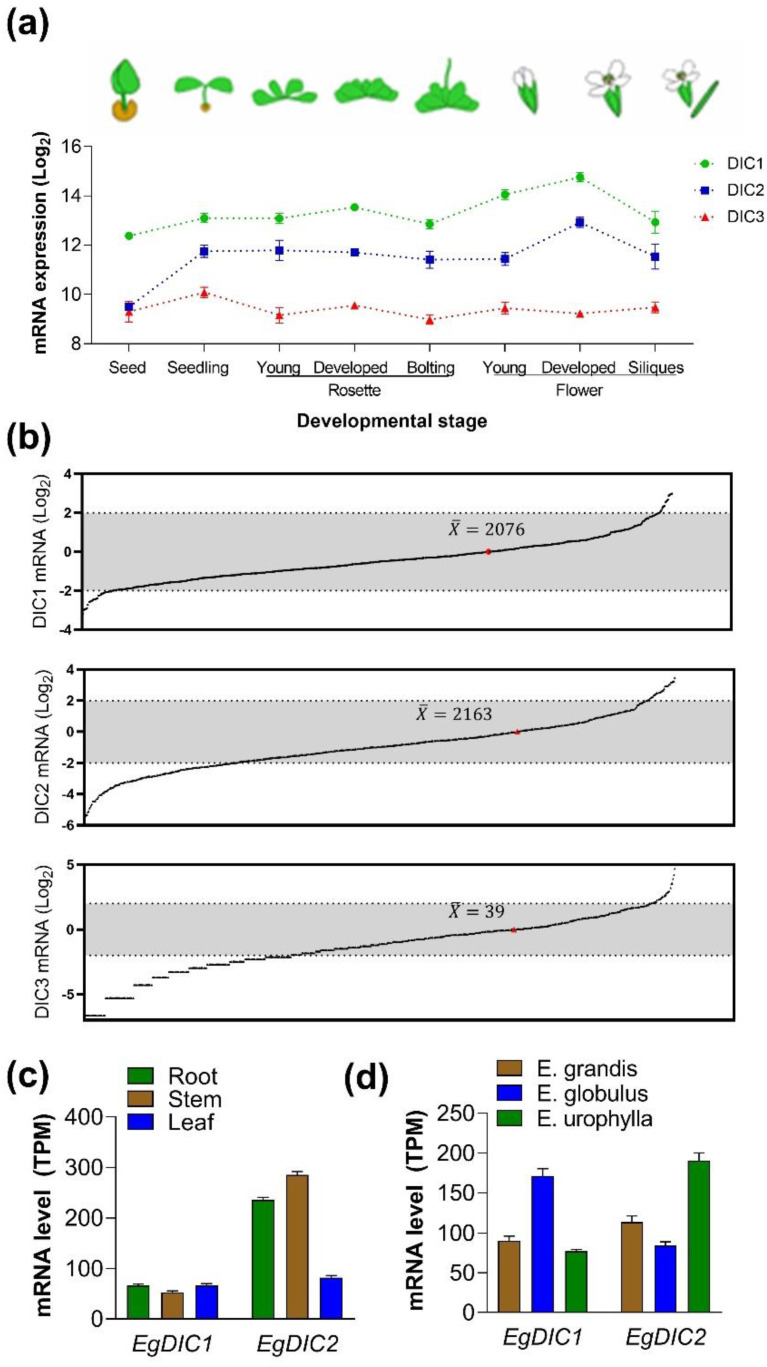
Developmental and natural variation in *DIC* gene expression. (**a**) Expression of the *AtDIC* family members along *A. thaliana* development as displayed by Genevestigator. (**b**) *AtDIC1-3* natural variation in expression across different *A. thaliana* ecotypes. X is the average expression across the ecotypes. Data points are fold-changes (Log_2_) calculated based on the average expression among accessions. Expression of *EgDIC1* and *EgDIC2* in distinct organs/tissues of *E. grandis* (**c**) and in the xylem of different *Eucalyptus* species (**d**). Data was obtained from the EUCANEXT database. TPM: Transcripts Per Kilobase Million.

**Figure 4 plants-11-00181-f004:**
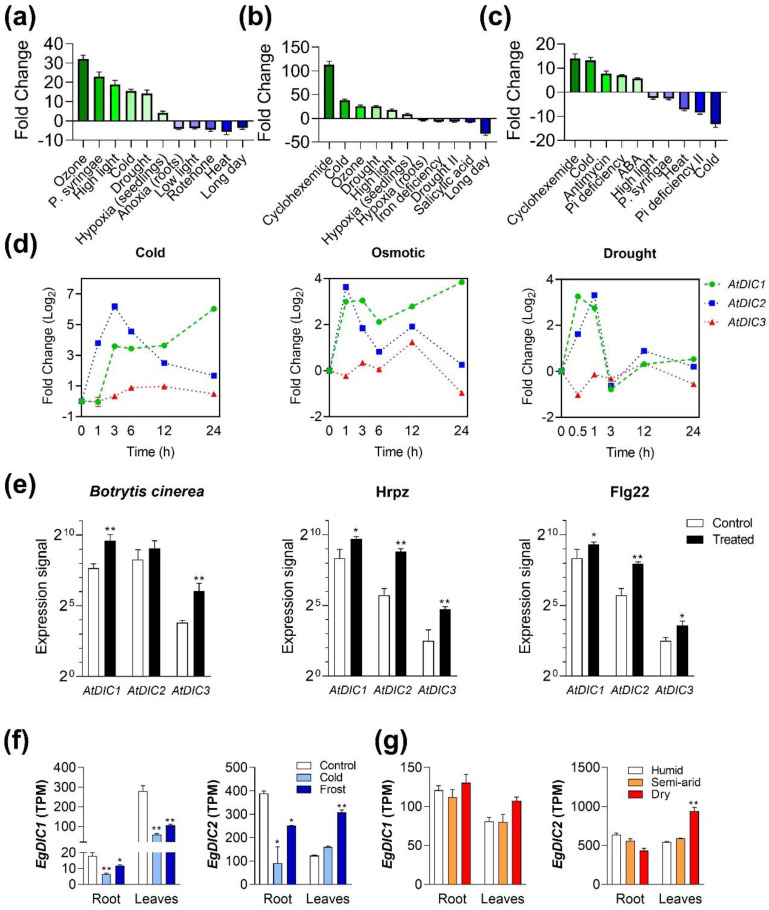
*Arabidopsis thaliana* DICs are responsive to abiotic and biotic stresses. (**a**) *AtDIC1* (**b**)**,**
*AtDIC2* and (**c**) *AtDIC3* expression in response to a wide variety of perturbations as displayed by Genevestigator. Data were filtered for *p* < 0.01. Temporal variation in *AtDICs* expression in plants subjected to (**d**) abiotic and (**e**) biotic stresses according to BAR. Expression of *EgDIC1* and *EgDIC2* in roots and leaves from eucalyptus subjected to (**f**) cold or frost, and (**g**) to different environmental conditions. Data was retrieved from the EUCANEXT database. Asterisks denote significant differences (* *p* <  0.1 and ** *p* <  0.05) compared with respective control (white bars).

**Figure 5 plants-11-00181-f005:**
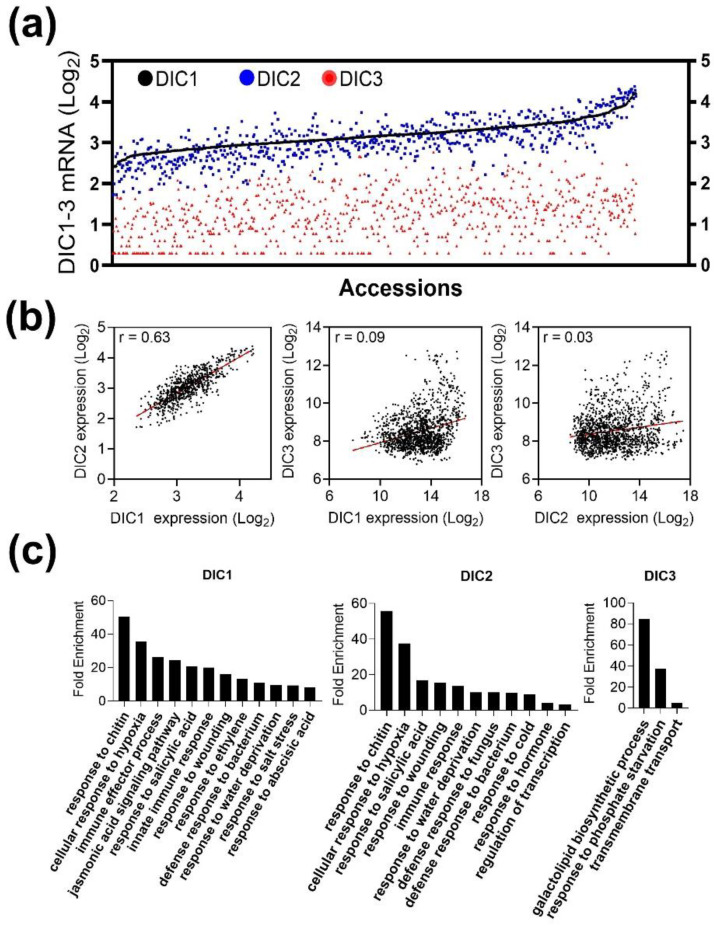
Expression of *AtDIC1* and *AtDIC2* is highly correlated across *Arabidopsis thaliana* ecotypes and in response to stresses. (**a**) Expression of the *AtDIC* family members across *A. thaliana* ecotypes. (**b**) Linear regression of *AtDIC1* and *AtDIC2* expression across *A. thaliana* ecotypes. (**c**) Gene ontology enrichment analysis of genes co-expressed with *AtDIC1, 2* and *3* across *A. thaliana* ecotypes.

**Figure 6 plants-11-00181-f006:**
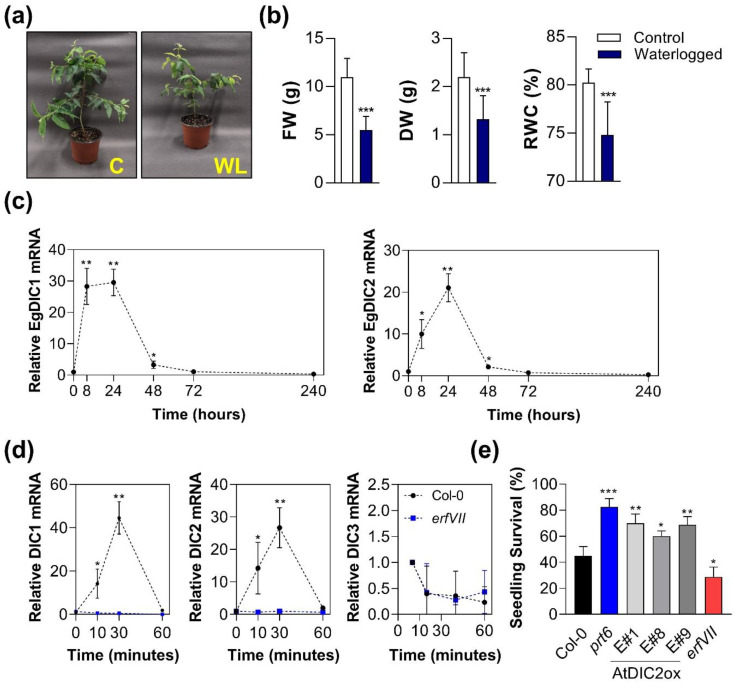
DICs are components of the low-oxygen response under control of the *ERFVIIs* transcription factors. (**a**) *E. grandis* seedlings (2-month-old) were subjected to 30 days of waterlogging (a representative image is shown). After 30 days, the aboveground parts of control (C) and waterlogged (WL) plants were used to determine (**b**) FW, DW and RWC. (**c**) Time course of *EgDIC1* and *EgDIC2* expression in roots after 0, 8, 24, 48, 72 and 240 h of waterlogging. (**d**) Time course of *AtDIC1*, *AtDIC2* and *AtDIC3* expression in Col-0 and *erfVII* seedlings after 0, 10, 30 and 60 min of submergence. (**e**) Arabidopsis seedling survival to 1 h submergence. E#1, E#8 and E#9—*AtDIC2* overexpressing lines; *prt6*—constitutive hypoxia response mutant; *erfVII*—defective hypoxia response mutant. Asterisks denote significant differences (* *p* < 0.1, ** *p* < 0.05 and *** *p* < 0.001) compared with Col-0 at 0 h.

## Data Availability

The data underlying this article are available within the article and in its online Appendix A.

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
