# Peer review of "Comprehensive In Silico Analysis and Transcriptional Profiles Highlight the Importance of Mitochondrial Dicarboxylate Carriers (DICs) on Hypoxia Response in Both Arabidopsis thaliana and Eucalyptus grandis"

_plants, 2022, doi:10.3390/plants11020181_

Round 1

Reviewer 1 Report

Authors have addressed most of my previous comments and hence can be accepted for publication

Author Response

We thank the reviewer for the positive evaluation of our revised version.

Reviewer 2 Report

In this re-submitted manuscript, improvements were made by the authors. However, the major concern for this manuscript: lack of novel data for DIC in plant oxygen-sensing in Arabidopsis and E. grandis still existed. Most of the data was still from published databases for DIC expression patterns in different developmental stages and in response to different abiotic and biotic stresses. In this case, most of the figures, to me, would rather be considered as preliminary data which was not enough to constitute a manuscript.

Author Response

We understand the reviewer concern but we still believe that novel and insightful results were generated using the erfvii mutant line and transgenic plants overexpressing AtDIC2. Moreover, the experimental data generated using eucalyptus is original and interesting for those working with forest trees. Thus, we kindly ask the reviewer to reconsider this point for manuscript acceptance.

Reviewer 3 Report

The authors debate on the role of mithocondrial DICs performing in silico end in vivo experiments. They use correct approach and discuss in appropriate way the results. The methodology is correct. The involvment of DICs in stress response may encourage others or the same authors to investigate on the argument. In my opinion the manuscript is suitable  fo publication.

Author Response

(The authors gave the same response as above.)

Reviewer 4 Report

The manuscript by Barreto P. et al. “Comprehensive in silico analysis and transcriptional profiles highlight the importance of mitochondrial dicarboxylate carriers (DICs) in plant oxygen-sensing in Arabidopsis thaliana and Eucalyptus grandis” provides new interesting data on the role of plant DICs under abiotic and biotic environmental challenges including submergence stress. Authors provided data of in silico and qRT-PCR expression analyses convincingly showing upregulation of DIC genes by pathogens, heat, cold, drought and hypoxia. The manuscript would merit publishing in Plants, but contains some inaccuracies and errors throughout the text.

1) The title is incorrect, since authors did not show participation of DICs in real oxygen-sensing or signaling events. DICs are carriers, not sensors. Therefore the title has to be changed to avoid mentioning of sensing.

2) Again, mitochondrial dicarboxylate carriers are just transporters, not enzymes; thus they cannot “catalyze” as is written in line 45.

3) According to the author’s guidelines, manuscript in MDPI journals should comprise separate “Results” and “Discussion” sections. Therefore, the manuscript has to be reorganized in a proper way.

4) Why did authors use amino acid sequences for alignment studies, but not protein-coding nucleotide sequences? The last ones were still used for in silico expression analysis and the search for regulatory elements. Moreover, due to amino acid alignments the gene names (in Italics) cannot be used (lines 94-102), these are proteins and have to be in regular font.

5) The same transcription factors (a-e) and genes (f-i) are presented in different colors at Figure 2. It has to be fixed; one TF or one gene must be in one color. For example, if AtDIC2 marked in red at Figure 2i, it should be red previously (f-h). Genes have to be in Italic in the figure (2i). Same should be employed for Figure 4 (a-c).

6) Why were FW, DW and RWC measured in aboveground part of Arabidopsis and gene expression – in roots?

7) Figure 6e capture contains no information about prt6 and E#1-9. It needs to be explained.

8) At lines 297-298 it is written “It should be emphasized that, according to our results, AtDIC2 is responsive to submergence in an ERFVII-dependent manner (Figure 6d).” Same can be seen for AtDIC1.

9) Conversion of malate to oxaloacetate is oxidation, not reduction as is written in line 308.

10) The origin of three independent transgenic lines (T2: E#1, E#8 and E#9) overexpressing AtDIC2 is not specified. If authors made transformation by themselves, the protocol description is required; if they got it from elsewhere it has to be mentioned either.

Taking all mentioned above into account, the manuscript requires major revision.

Author Response

1) The title is incorrect, since authors did not show participation of DICs in real oxygen-sensing or signaling events. DICs are carriers, not sensors. Therefore the title has to be changed to avoid mentioning of sensing.

R: Following the reviewer’s suggestion, the title was changed to “Comprehensive in silico analysis and transcriptional profiles highlight the importance of mitochondrial dicarboxylate carriers (DICs) on hypoxia response in both Arabidopsis thaliana and Eucalyptus grandis”.

2) Again, mitochondrial dicarboxylate carriers are just transporters, not enzymes; thus they cannot “catalyze” as is written in line 45.

R: We thank the reviewer for pointing this out. This was corrected in the revised version.

3) According to the author’s guidelines, manuscript in MDPI journals should comprise separate “Results” and “Discussion” sections. Therefore, the manuscript has to be reorganized in a proper way.

R: According to the Plants author’s guidelines, the results section may be combined with discussion. This information can be found at the MDPI website in “Author Guidelines” at the “Discussion” subtopic (https://www.mdpi.com/journal/plants/instructions).

4) Why did authors use amino acid sequences for alignment studies, but not protein-coding nucleotide sequences? The last ones were still used for in silico expression analysis and the search for regulatory elements. Moreover, due to amino acid alignments the gene names (in Italics) cannot be used (lines 94-102), these are proteins and have to be in regular font.

R: Amino acid sequences were used in our alignment studies because it is more appropriate when comparing sequences from species that are phylogenetically distant such as Arabidopsis thaliana and Amborella trichopoda. As pointed out by the reviewer, the names at lines 94-102 were changed to regular font instead of italic.

5) The same transcription factors (a-e) and genes (f-i) are presented in different colors at Figure 2. It has to be fixed; one TF or one gene must be in one color. For example, if AtDIC2 marked in red at Figure 2i, it should be red previously (f-h). Genes have to be in Italic in the figure (2i). Same should be employed for Figure 4 (a-c).

R: Color scheme was changed in Figure 2 to follow the reviewer's recommendation. The names of the genes depicted in Figure 4 were changed to italic.

6) Why were FW, DW and RWC measured in aboveground part of Arabidopsis and gene expression – in roots?

R: The reviewer probably refers to Eucalyptus grandis and not to Arabidopsis thaliana in this query. Aboveground part of Eucalyptus grandis was used for weight measurements because it is of interest to the pulp and paper companies to explore the aboveground part of this species. Therefore, we sought to investigate the submergence effect on the aerial part of the plants.  On the other hand, gene expression analysis was conducted in roots because it was previously reported that waterlogging in trees has a more prominent impact on root gene expression when compared to leaves (please see Ref. nº 45).

7) Figure 6e capture contains no information about prt6 and E#1-9. It needs to be explained.

R: We thank the reviewer for pointing this out. The caption of Figure 6e was amended to include this missing information.

8) At lines 297-298 it is written “It should be emphasized that, according to our results, AtDIC2 is responsive to submergence in an ERFVII-dependent manner (Figure 6d).” Same can be seen for AtDIC1.

R: The reviewer is correct. We modified this sentence in the revised version to include this information.

9) Conversion of malate to oxaloacetate is oxidation, not reduction as is written in line 308.

R: We thank the reviewer for pointing this out. This was corrected in the revised version.

10) The origin of three independent transgenic lines (T2: E#1, E#8 and E#9) overexpressing AtDIC2 is not specified. If authors made transformation by themselves, the protocol description is required; if they got it from elsewhere it has to be mentioned either.

R: We thank the reviewer for pointing this out. In fact, these transgenic lines were generated in house. The missing information regarding the methods used to generate these lines is now included in the corresponding section of the revised version.

Reviewer 5 Report

The authors take on an important challenge to investigate the expression of Arabidopsis thaliana DIC family and their homologues in Eucalyptus grandis. The authors also suggest that these proteins have an important function in the adaptation to environmental stresses and part of a low-oxygen response in both species. Although I think the authors make here an important contribution, I have some comments concerning certain statements and data presentation: 

Major comments:

  • Line 173. The authors mention that they “searched public transcriptomic datasets using Genevestigator software to investigate the expression patterns of thaliana DICs along plant development”. Are there particular tissues or developmental stages enriched in DIC expression?
  • Line 178-186. The authors say that “AtDIC3 expression remains very low in all developmental stages, a finding that led to a previous suspicion that this gene could be a pseudogene”. I’m not sure how this conclusion could be made only based on Genevestigator. Please elaborate. Could it be that DIC3 is more active in specific tissues and low expression is detected in global dataset due to its restricted and specific action?
  • Line 184. “Interestingly, AtDIC3 expression was absent in at least 25 of the investigated accessions and its highest expression value was 1003, less than half of the detected average expression of AtDIC1 and AtDIC2”. This relates to the previous comment. Could it be that DIC3 is expressed and required in fewer cells and has a specific role only in these cells and a certain developmental stage only?
  • Line 198. “We could also observe that EgDIC1 expression was almost 2 times higher in the xylem of globulus….” What was the reasoning here for looking at xylem and not at other tissues? this relates to the previous questions related to specificity of the expression of different DICs.
  • Figure 2a-e and line 119. “DIC promoters binding sites are enriched for stress-responsive transcription factors”. The authors show the binding sites of large families of transcription factors and not all of them are related to the stress response. Please clarify the statements here.
  • Figure 2f. Please show which transcription factors are co-expressed between DIC1 and 2 and individual DICs. Could these co-expression patterns be influenced by distinct tissue-specific patterns of the expression of different DICs?

Minor comments:

  • Line 116. “…..light, abscisic acid (ABA) and hypoxia are common elements among the promoter regions….” these are not elements of the promoter, correct accordingly.
  • Line 175. “Subsequently, as plants go through the seedling stage, AtDIC2 is markedly induced”. The description is very vague, what does it mean markedly induced? where?
  • Line 301. Elaborate what is the line prt6.
  • Line 354. what does “After 4 days, the root tips were scored based on growth” mean? it’s a vague description.
  • Line 383. “Results from Real-time PCR was analyzed…” correct the sentence.

Author Response

Major comments:

  • Line 173. The authors mention that they “searched public transcriptomic datasets using Genevestigator software to investigate the expression patterns of thaliana DICs along plant development”. Are there particular tissues or developmental stages enriched in DIC expression?

R: According to the Genevestigator database, AtDIC1 and AtDIC2 are almost ubiquitous while AtDIC3 expression is not enriched in any particular tissue or developmental stage. On the other hand, AtDIC3 expression is altered when plants are subjected to some perturbations (Figure 4c).

  • Line 178-186. The authors say that “AtDIC3 expression remains very low in all developmental stages, a finding that led to a previous suspicion that this gene could be a pseudogene”. I’m not sure how this conclusion could be made only based on Genevestigator. Please elaborate. Could it be that DIC3 is more active in specific tissues and low expression is detected in global dataset due to its restricted and specific action?

R: We thank the reviewer for pointing this out. We agree with the reviewer that this conclusion could not be made using exclusively Genevestigator data. This hypothesis was raised previously and already disproven (please see Ref. nº. 3). This was thought to be due to the low expression of AtDIC3, which is probably associated with the facts mentioned by the reviewer. As other reviewers have also criticized this sentence, we decided to delete it from the revised version.

  • Line 184. “Interestingly, AtDIC3 expression was absent in at least 25 of the investigated accessions and its highest expression value was 1003, less than half of the detected average expression of AtDIC1 and AtDIC2”. This relates to the previous comment. Could it be that DIC3 is expressed and required in fewer cells and has a specific role only in these cells and a certain developmental stage only?

R: We agree with the referee and because of that, the following sentence was added in the revised version of the manuscript: “It seems therefore that AtDIC3 activity is required under specific circumstances and at low levels.”

  • Line 198. “We could also observe that EgDIC1 expression was almost 2 times higher in the xylem of globulus….” What was the reasoning here for looking at xylem and not at other tissues? this relates to the previous questions related to specificity of the expression of different DICs.

R: Xylem tissue is of particular interest for E. grandis breeding, especially in programs aiming to improve fiber characteristics. For this reason, xylem data is frequently available in public databases. On the other hand, as it is mentioned in the manuscript, the availability of public datasets from E. grandis organs/tissues and stress conditions is very limited, a feature that restricts the search of gene expression in this species.

  • Figure 2a-e and line 119. “DIC promoters binding sites are enriched for stress-responsive transcription factors”. The authors show the binding sites of large families of transcription factors and not all of them are related to the stress response. Please clarify the statements here.

R: We agree with the referee that the figure title was not in accordance with what is depicted in Figure 2. Therefore, the title was rephrased to “Promoter and co-expression analysis reveal transcription factors that are potential regulators of AtDICs expression.” 

  • Figure 2f. Please show which transcription factors are co-expressed between DIC1 and 2 and individual DICs. Could these co-expression patterns be influenced by distinct tissue-specific patterns of the expression of different DICs?

R: The transcription factors that were found co-expressed with individual DICs were already provided (please see Table S5). We have now included lists of transcription factors co-expressed among AtDICs (Table S6) in the revised version. 

Although we agree with the referee, we believe that the co-expression patterns observed in our research provide a good inference of what is found during plant development. It is important to emphasize that the mentioned data was collected from more than 700 distinct ecotypes and combined with Genevestigator datasets.

Minor comments: 

  • Line 116. “…..light, abscisic acid (ABA) and hypoxia are common elements among the promoter regions….” these are not elements of the promoter, correct accordingly.

R: We thank the reviewer for pointing this out. This sentence was corrected in the revised version of the manuscript.

  • Line 175. “Subsequently, as plants go through the seedling stage, AtDIC2 is markedly induced”. The description is very vague, what does it mean markedly induced? where?

R: We improved this sentence in the revised manuscript in order to make the statement more precise.

  • Line 301. Elaborate what is the line prt6.

R: Additional information was included in the revised version to clarify this point.

  • Line 354. what does “After 4 days, the root tips were scored based on growth” mean? it’s a vague description.

R: We improved the description of this methodology as requested.

  • Line 383. “Results from Real-time PCR was analyzed…” correct the sentence.

R: This typo was corrected accordingly.

Round 2

Reviewer 4 Report

The authors took into account most of the comments of the reviewer. I see no obstacles to the publication of the manuscript.